# Improving prediction of cervical cancer using KNN imputer and multi-model ensemble learning

**Turki Aljrees** *

College of Computer Science and Engineering, University of Hafr Al-Batin, Hafar Al-Batin, Saudi Arabia

* tajrees@uhb.edu.sa

## Abstract

Cervical cancer is a leading cause of women's mortality, emphasizing the need for early diagnosis and effective treatment. In line with the imperative of early intervention, the automated identification of cervical cancer has emerged as a promising avenue, leveraging machine learning techniques to enhance both the speed and accuracy of diagnosis. However, an inherent challenge in the development of these automated systems is the presence of missing values in the datasets commonly used for cervical cancer detection. Missing data can significantly impact the performance of machine learning models, potentially leading to inaccurate or unreliable results. This study addresses a critical challenge in automated cervical cancer identification—handling missing data in datasets. The study present a novel approach that combines three machine learning models into a stacked ensemble voting classifier, complemented by the use of a KNN Imputer to manage missing values. The proposed model achieves remarkable results with an accuracy of 0.9941, precision of 0.98, recall of 0.96, and an F1 score of 0.97. This study examines three distinct scenarios: one involving the deletion of missing values, another utilizing KNN imputation, and a third employing PCA for imputing missing values. This research has significant implications for the medical field, offering medical experts a powerful tool for more accurate cervical cancer therapy and enhancing the overall effectiveness of testing procedures. By addressing missing data challenges and achieving high accuracy, this work represents a valuable contribution to cervical cancer detection, ultimately aiming to reduce the impact of this disease on women's health and healthcare systems.

## Introduction

Cervical cancer develops in the cervix cells of the uterus's lower part. Cervical cancer is usually caused by infection with human papillomaviruses (HPV), which is spread through sexual contact. Cervical cancer is the second largest cause of cancer mortality among women, according to statistics [1]. The majority of the documented cases in 2018 are from poor and developed countries [2]. Cervical cancer accounts for 6.5 per cent of all cancers in women. According to predictions, there will be around 342,000 cervical cancer deaths and 604,127 new instances of the disease worldwide in 2020 [3]. Cervical cancer is a major public health concern,

**Data Availability Statement:** The dataset is publicly available on the link (https://www.kaggle.com/datasets/ranzeet013/cervical-cancer-dataset) and its reference also added in the dataset section.

**Funding:** The author received no specific funding for this work. He will manage the APC himself.

**Competing interests:** The authors have declared that no competing interests exist.

particularly in wealthy countries: Every year, 54,517 newly diagnosed cases of invasive cervical cancer are recorded in Europe, and 24,874 women die due to cervical malignancies [4]. Vaccinations are frequently included in routine immunization programs. By 2030, it is projected that 90% of European girls would have gotten the complete HPV vaccination by the age of 15 [5]. Furthermore, the WHO urges European decision-makers to step up efforts to eliminate cervical cancer by using current preventative tools [6]. Regular examinations and early identification, like with most other diseases, can considerably lower the risk of death [7]. Due to the absence of symptoms in the early stages of the disease, cervical cancer detection might be difficult. Regular yearly check-ups frequently reveal changes to the cervix. The most prevalent symptom of cervical cancer is irregular bleeding, which can vary depending on the stage of the disease. In about 90% of instances, the late stages of the disease show different indicators [8]. Spotting, contact bleeding, postmenopausal bleeding, and brownish discharge are all typical symptoms of this illness. In addition to the above-described symptoms, the presence of bloody discharge—which typically has an unpleasant odour is another vital sign of cervical cancer. Lower abdominal discomfort may arise as a result of surrounding organ involvement when the illness has progressed to an advanced stage [9].

Cervical cancer is a major public health concern. Since the middle of the past century, developed countries have introduced cytology-based screening programs as PaP smear tests [10]. The PaP smear test is a screening process used to detect the presence of malignant or precancerous cells in the cervix (uterine opening). During this procedure, a sample of the cervix is softly scraped and analyzed to determine the proliferation of abnormal cells. Computer-aided detection (CAD) approaches are becoming increasingly popular as an alternative to manual diagnosis since they are less susceptible to human errors [11]. Repeated screening is required due to the low sensitivity of cytology [12]. Several research publications on the early identification of cervical cancer using machine learning and deep learning have been published in the last few years [13]. These algorithms are often trained on a specific dataset to extract certain characteristics for classification.

The potential of electronic health records (EHRs) to offer insightful data on illness progression can result in better healthcare [14]. As a result, a variety of useful uses for EHRs, including disease risk assessment, adverse event identification, and medical decision support, are gaining attention [15]. Mining EHRs has a tremendous potential to provide novel clinical hypotheses by observing big populations over time and looking at the relationship between clinical events and outcomes [16]. The goal of this study is to use EHRs as a source of data for the purpose of detecting cervical cancer, the most common malignancy in women. After breast, colon, and lung cancer, cervical cancer is the fourth most frequent illness in women, with over 500,000 women receiving a diagnosis each year. In Europe, there are between 12 and 30 instances of cervical cancer per 100,000 females [17]. The frequency of cervical cancer has steadily declined over the years thanks to early detection of the disease and preceding changes. Cervical cancer used to make up to 70% of all genital cancers, but it now only makes up 35—50% of cases, with a trend to decline further. The age range most affected by cervical cancer is between 20 and 40 years old, while among women over 40 years old, the incidence declines; just 1-2 per cent of these women are diagnosed with cervical cancer [18].

Machine Learning allows computers to learn from historical or previous experiences, assisting in the prediction of current scenarios based on complex data in cervical cancer diagnosis [19]. Machine Learning employs a variety of statistical, and optimisation approaches to determine if an instance is malignant or non-cancerous. Authors proposed a stacked ensemble model to predict cervical cancer [20]. Authors applied various machine learning models including RF, SVM and XGBoost in combination with wrapper techniques for feature

selection to improve the diagnosis of cervical cancer [21]. In another recent work [22], authors applied transfer learning models to predict cervical cancer dose.

Many individuals are unaware of the risks associated with cervical cancer, a serious condition that can be deadly. Early diagnosis and treatment, however, can aid in its prevention [9]. Cervical cancer is a major public health concern, especially in regions with high incidence rates. Existing screening methods have limitations, and there is an urgent need to improve cervical cancer diagnosis. This study explores the potential of EHRs and machine learning to develop more accurate and efficient diagnostic tools. Unfortunately, there are many countries without reliable screening techniques for finding this kind of cancer. Existing data sets are not suitable for machine learning models because they may contain missing or null values, which have a significant influence on the effectiveness of models. The research in cervical cancer detection aims to address a critical gap in existing methods, primarily related to the handling of missing data in machine learning approaches. While acknowledging the significance of early and accurate cervical cancer detection, the paper highlights the limitations of traditional methods and emphasizes the potential of machine learning. The research gap is identified as the lack of comprehensive studies and robust methodologies for handling missing data in cervical cancer datasets. This paper explores two missing values imputer techniques to resolve this problem. Also suggested is an ensemble model to improve cervical cancer diagnosis accuracy. In contrast to the suggested method, we also assess the efficacy of several machine-learning approaches for cervical cancer diagnosis.

The following are the **main** contributions made by this research endeavor:

- A unique ensemble model is put forth in this work to forecast cervical cancer in patients. Extreme gradient boosting (XGB), random forest (RF), and extra tree classifier (ETC) are the foundations of the proposed ensemble model, and a voting mechanism is used to determine the final prediction.

- The KNN (K closest neighbour) imputer is used in studies to produce missing values to address the issue of missing values.

- The Principal Component Analysis (PCA) is utilized to impute missing values on considering feature importance.

- Different machine learning models, including RF, LR (logistic regression), GBM (gradient boosting machine), GNB (Gaussian Naive Bayes), ETC, SVC (support vector classifier), DT (decision tree), and SGD (stochastic gradient descent), are used to compare their performances. The performance of the suggested model is compared against cutting-edge methods in terms of accuracy, precision, recall, and F1 score to assess its efficacy.

The remainder of the paper is divided into the following sections. A survey of the pertinent literature on the use of machine learning algorithms in the diagnosis of cervical cancer is included in Section. A summary of the dataset and the machine learning models utilized in the study can be found in Section. It also goes into great length about the suggested process. The study's findings are presented in section which also explores their ramifications. Section provides closing observations and makes possible directions for further investigation.

## Related work

There have been many different tactics used in the years-long fight against cancer. While total eradication might not be feasible, early detection and prognosis of the disease might help lessen its effects. Cervical cancer is difficult to identify in its early stages since it often has no early signs, even though early identification is essential for successful treatment. Therefore, the

only approach to detect the existence of malignant cells is through routine screening. Risks might arise because screening findings occasionally provide false-positive results or are delayed. Artificial intelligence (AI) has been implemented in the healthcare industry to address these issues. The accuracy and speed of predicting malignant cells are being increased while the number of false positives is decreased via the application of various algorithms, tools, and methodologies. Various methods have been employed by numerous studies to predict cervical cancer.

The classification of cervical cells in pap smear pictures is thoroughly studied by Yaman and Tuncer using two datasets: SIPaKMeD and Mendeley LBC [23]. The authors used SVM (support vector machine) for classification and NCA (neighbourhood component analysis) to pick 1000 characteristics. They used hold-out validation and five-fold cross-validation to validate the results (80:20). The Mendeley LBC and SIPaKMeD datasets showed high accuracy rates of 98% and 99%, respectively, in the findings. Similar to this, Das et al. [24] suggested a technique to identify cervical cancer by adjusting pre-trained deep network models to extract deep convolutional features, such as Inception ResNet, DenseNet-121, VGG19 Net, ResNet-50V2, and InceptionV3. To capture common visual traits, they also employed local binary patterns and an oriented gradient histogram. The findings of ResNet-50V2 and DenseNet-121 applied separately are found to be less accurate than the combination of deep features derived from DenseNet-121 and ResNet-50V2, which had an average classification accuracy of 95.33 per cent. They also noticed an improvement in recognition skills of 91%.

Alquran et al. [25] classified cervical cancer into seven categories using the Harvel dataset by combining deep learning (DL) with a cascade SVM classifier, with an accuracy of up to 92%. According to the study, combining traditional machine learning methods with deep learning may accurately classify cervical cancer. Using the UCI cervical cancer risk factor dataset, Alsmariy et al. in [26] developed a machine-learning algorithm to predict cervical cancer. They used PCA (principal component analysis) in conjunction with SMOTE (synthetic minority oversampling approach) to address the problem of class imbalance. The system scored a phenomenal 98% accuracy. Lilhore et al. [27] proposed an ensemble model to detect cervical cancer in the meanwhile. A feature selection method, prediction model, and SVM classifier are all part of the suggested methodology. An enhanced variant of random forest known as brutal analysis discovers subsets from the data source that significantly affect classification accuracy. Their study's conclusions showed that Boruta with SVM had an accuracy score of 0.912.

An automated technique called CervDetect is presented by Mavra Mehmood et al to identify cancerous cervical development. In essence, CervDetect is a machine learning-based system that pre-processes data using the Pearson correlation between the input and output variables. CervDetect employs the RF model for the important feature selection process. The study's findings indicate that CervDetect had an accuracy score of 93% and a mean square error of 0.07111. Anandaraj et al. [28] provide an overhead cross-section sampling machine learning model for cervical cancer prediction. The authors employed the oversampling technique SMOTE to address the issues of class imbalance. They employed thirteen machine learning models to evaluate the effectiveness of the suggested approach. The RF classifier scored 96% accurate on the unbalanced dataset, while the same classifier maintained a 98% accuracy rate when utilizing the oversampling strategy. Other medical domains also employed deep networks like on CT image sequences [29, 30], surgical instrument designing [31], EEG decoding [32], image analysis [33], and cancer diagnosis [34]. Feature-based networks are designed by researchers in tasks like object detection [35, 36], and bidding behaviour [37].

A machine-learning system for cervical cancer prediction is proposed in the paper of J.F. Kaleema [38]. Three machine learning models are utilized in the study: LR, DT, and XGBoost. The study's findings showed that when employing the chosen features, machine learning

classifiers performed at their best. A clever method of cervical cancer forecasting is put out by Mudawi and Alazeb [39]. They utilized six machine learning models, and the RF, DT, ADA, and GBM models all performed well in terms of accuracy on the characteristics they chose. Quinlan et al. [40] compared several machine-learning methods for classifying cervical cancer. An unbalanced dataset is employed, and it is addressed using the SMOTE-Tomek resampling approach and a tuned RF. According to the results, the RF with S-Tom had a 99% accuracy rate.

Utilizing improved feature selection approaches and classification methodologies, Nithya and Ilango [41] undertook research on the prediction of cervical cancer. The following five machine learning models are used: random forest, KNN, SVM, Rpart, and C5.0. After using feature selection techniques to pinpoint crucial characteristics, an improved feature selection is created. According to the study's findings, C5.0 fared better than other models in terms of accuracy score. A machine learning approach for cervical cancer prediction is also proposed by Gowri and Saranya in [42], with the goal of achieving high accuracy. They identified outliers in the dataset using DBSCAN and SMOTE-Tomek, and they made predictions using the two scenarios DBSCAN+SMOTE-Tomek+RF and DBSCAN+SMOTE+RF. The study found that DBSCAN+SMOTE+RF achieved a 99& accuracy value.

In recent times, machine learning and deep learning models have emerged as valuable solutions for disease diagnosis like shoulder segmentation [43], malarial parasite detection [44], tumour infiltrating [45], brain tumour classification [46, 47], COVID-19 diagnosis [48, 49], lymphocyte analysis [50], medical image analysis and Malarial parasite classification [51]. Researchers applied deep learning models to predict the survival rate of cervical cancer patients [52]. Authors applied deep learning models on the pap-smear dataset to classify cervical cancer [53]. Pretrained models like AlexNet, GoogleNet and ResNET are employed for feature extraction and machine learning models are used for classification using Pap smear dataset for cervical cancer prediction [54]. Recently, the CNN model and its variants like Transformers [55] have been applied in medical image analysis [56] and Lymphocyte detection cancer patients [57].

These papers show the effectiveness of machine learning approaches for identifying and forecasting cervical cancer, including deep learning and ensemble methods. They also emphasize the significance of correcting feature selection and class imbalance to raise the models' accuracy. Overall, these methods have the potential for raising cervical cancer detection and prognosis efficiency and accuracy.

## Materials and methods

In this part, we give a brief review of the data set that is utilized, the data preparation methods that are used, the machine learning algorithms that are used to detect cervical cancer and a synopsis of the class-balancing methods that are applied in this work.

### Dataset

This study made use of a publicly accessible data collection from the Hospital Universitario de Caracas in Venezuela called the [58]. The only publicly accessible data collection at this time that can be used to develop AI algorithms and questionnaires for a future cervical cancer screening survey is this one. The researchers wanted to determine if AI models and class-balancing methods are appropriate for creating such a study. The data set, which consists of a total of 858 occurrences and 36 characteristics, is broken down into 35 input variables and one output variable in the Table 1. Table 1 has a detailed description of each input variable.

**Table 1. Description of the dataset used in this study.**

| Number | Attribute name | Type | Range | Missing Values |
|---|---|---|---|---|
| 1 | Age | int | 13-84 | 0 |
| 2 | IUD (Years) | int | 0-19 | 117 |
| 3 | STDs: genital herpes | bool | 0-1 | 105 |
| 4 | Hormonal contraceptives | bool | 0-1 | 108 |
| 5 | Dx: cancer | Bool | 0-1 | 0 |
| 6 | Smokes | Bool | '0-1 | 13 |
| 7 | STDs: vaginal condylomatosis | Bool | 0-1 | 105 |
| 8 | STDs: AIDS | Bool | 0-1 | 105 |
| 9 | Num of Pregnancies | Int | 0-110 | 56 |
| 10 | Intrauterine Device (IUD) | Bool | 0-1 | 117 |
| 11 | STDs: cervical condylomatosis | Bool | 0-1 | 105 |
| 12 | STDs: molluscum contagiosum | Bool | 0-1 | 105 |
| 13 | STDs: time since last diagnosis | Int | 0-3 | 787 |
| 14 | Cytology | Bool | 0-1 | 0 |
| 15 | First sex intercourse(age) | Int | 10-32 | 7 |
| 16 | Hormonal contraceptives (years) | Int | 0-22 | 108 |
| 17 | STDs: condylomatosis | Bool | 0-1 | 105 |
| 18 | STDs: Time since first diagnosis | Int | 0-1 | 787 |
| 19 | Schiller | Bool | 0-1 | 0 |
| 20 | Number of sexual partners | Int | 1-28 | 26 |
| 21 | Smokes (packs/year) | int | 0-37 | 13 |
| 22 | STDs (number) | Int | 0-4 | 105 |
| 23 | STDs:pelvic inflammatory disease | Bool | 0-1 | 105 |
| 24 | STDs: Number of diagnosis | Int | 0-1 | 0 |
| 25 | Hinselmann | Bool | 0-1 | 0 |
| 26 | Diagnosis:Dx | Bool | 0-1 | 0 |
| 27 | STDs: Hepatitis B | Bool | 0-1 | 105 |
| 28 | Smokes (years) | int | 0-37 | 13 |
| 29 | Sexually Transmitted Disease (STD) | Bool | 0-1 | 105 |
| 30 | STDs: syphilis | Bool | 0-1 | 105 |
| 31 | Dx: Human Papillomavirus (HPV) | Bool | 0-1 | 0 |
| 32 | STDs: vulva-perineal condylomatosis | Bool | 0-1 | 105 |
| 33 | STDs: HPV | Bool | 0-1 | 105 |
| 34 | Dx: cervical intraepithelial Neoplasia (CIN) | Bool | 0-1 | 0 |
| 35 | STDs: HIV | Bool | 0-1 | 105 |
| 36 | Biopsy (target Variable) | bool | 0-1 | |

The data set clearly displays a serious class imbalance, as shown in Table 1. The kNN imputer is used in this work as a data balancing tool because of the challenges that come with the categorization of unbalanced data.

## Data preprocessing

Data preprocessing, which entails deleting redundant or unnecessary data from the dataset to increase model efficacy and save computation time, is crucial for achieving optimal performance from machine learning models. The dataset's significant number of missing values, which are categorized by class in Table 1, are found during the research's data preparation

stage. There are a sizable number of missing values in the dataset, as shown by Table 1. Since the dataset's data are categorical, there are three methods for handling the missing values.

- Deleting all instances of the missing values from the dataset.

- using the KNN imputer method.

- Employing PCA method to pick important features.

**Deleting all instances of the missing values.** Removing any instances that have missing values is one method for addressing the dataset's missing values. We choose to test this strategy in the first experiment by eliminating all fields with blank values.

**Using KNN imputer method.** The KNN imputer may be used to address missing values in the data for the second choice to handle them. In the modern world, data is gathered from many different sources and used for a variety of things, such as analysis, insightful insights, and hypothesis confirmation. Handling missing values is a crucial preprocessing step since mistakes in data extraction or collection might lead to missing data. The imputation technique you choose is important because it may have a big influence on how well your model performs. A popular approach for imputing missing data is the KNN imputer, which may be used instead of more conventional imputation techniques like [59]. By utilizing the Euclidean distance matrix to locate the closest neighbors, it substitutes missing values. By discarding the missing values and assigning more weight to the non-missing coordinates, the Euclidean distance is determined. Euclidean distance may be calculated in order to determine distance as

$$D_{xy} = \sqrt{weight*squared\ distance\ from\ present\ coordinates} \tag{1}$$

where

$$weight = \frac{total\ number\ of\ coordinates}{number\ of\ present\ coordinates} \tag{2}$$

**PCA method.** Assuming that the data has a multivariate normal distribution and that the missing values are absent at random is one technique to use PCA to impute missing data. This indicates that neither a value's value nor any other variable affects the likelihood that a value will be absent [60]. PCA provides various benefits over alternative approaches for imputed missing data, such as eliminating instances with missing values, employing mean or median imputation, or utilizing regression techniques. These benefits include preserving variable relationships and correlations by using the covariance matrix to estimate missing values, reducing data dimensionality to improve the performance and efficiency of subsequent analyses, and handling large amounts of missing values. Furthermore, because it just requires a linear transformation of the data, PCA is simple to execute and comprehend.

PCA imputation techniques for filling in missing values in a dataset involve the following steps:

1. Data Preparation: Begin with a dataset that contains missing values. These missing values exist in more than one column or feature of the dataset.

2. Data Standardization: Standardize the dataset by centering it (subtracting the mean) and scaling it (dividing by the standard deviation). This step ensures that all features have similar scales, which is important for PCA.

3. PCA: Apply Principal Component Analysis to the dataset. PCA identifies the principal components (linear combinations of the original features) that capture the most variance in the data. These principal components are ordered by their ability to explain variance, with the first component explaining the most variance, the second component explaining the second most, and so on.

4. Imputation: For missing values in the dataset, we use the information from the PCA-transformed data to impute those missing values. This typically involves projecting the data with missing values onto the PCA space and then back-transforming it to the original feature space. The imputed values are obtained by using the coordinates of the projected data on the principal components.

PCA imputation is based on the assumption that the variance in the data captures meaningful patterns, and it leverages this variance to estimate missing values. It can be a useful technique when dealing with datasets with missing values, especially when those missing values are missing at random like in this case.

## Machine learning classifiers

This section discusses the machine-learning algorithms used to identify cervical cancer, using Python to develop all eight of the supervised machine-learning algorithms. These methods are often used to solve classification and regression issues. The efficiency of the proposed system is assessed using tree-based algorithms, regression-based models, and ensemble models. In this work, the classification problem is tackled using a total of eight different machine learning methods, as well as an ensemble learning model presented in Table 2.

## Proposed approach

A well-known source of publicly accessible data sets, Kaggle provided the dataset used in this study. Preprocessing is done to deal with the problem of missing values and enhance the functionality of the learning models. KNN imputer is used to handle the missing data. The data is then divided into two proportions: 70% for model training and 30% for model testing. The XGB+RF+ETC ensemble technique is utilized by the proposed cervical cancer detection system. A strong method for integrating the predictions of many models to increase accuracy and resilience is called an ensemble model. Higher overall performance can be obtained by integrating the strengths and shortcomings of each model in an ensemble. This work suggests a methodology of ensemble learning for detecting cervical cancer that includes the well-known XGB, RF, and ETC algorithms. Fig 1 displays the suggested approach's workflow diagram.

Three independent machine learning algorithms' predictions are combined in the ensemble model to produce results. Predictions from several models that are trained on the same dataset are combined to create an ensemble model. This strategy is used by the XGB+RF+ETC ensemble model, which trains the XGB, RF, and ETC models independently on the same dataset. Predicted probabilities are generated for each class of the target variable by each of these models. The final forecast for each observation in the dataset may be created by combining these anticipated probabilities. Taking a weighted average of the projected probabilities, with the weights based on how each model performed on a validation set, is a typical method of combining the predictions.

The proposed model makes predictions that are more reliable and accurate by integrating the strengths of three separate machine-learning algorithms. We may enhance the generalization performance of the model and lessen overfitting by training multiple models on the cervical cancer dataset, and then merging their predictions. The suggested ensemble model's

**Table 2. Machine learning models.**

| Reference | Model | Description |
|---|---|---|
| [61] | RF | The RF classifier is a decision tree-based system that employs numerous weak learners to make very accurate predictions. RF uses a technique known as 'bootstrap bagging' to train multiple decision trees using various bootstrap samples. A bootstrap sample is generated by randomly subsampling the training dataset, which has the same size as the test dataset. RF, like other ensemble classifiers, makes predictions using decision trees. Choosing the root node at each level of decision tree construction might be difficult. |
| [62] | DT | DT is a well-known machine learning technique with several applications in regression and classification issues. At each level of the DT, picking the root node, also known as 'attribute selection,' is a vital element of the process. The 'Gini index' and 'information gain' are both well-known attribute selection approaches. The Gini index is often used to determine the extent of impurity in a dataset. |
| [63] | LR | Because of its use of the logistic equation (also known as the sigmoid function), LR is a popular approach for dealing with binary classification issues. This function converts every given numerical value into a number between 0 and 1 using an S-shaped curve, which is why LR is so popular. |
| [64] | SVC | The SVC is a well-known supervised machine learning technique for tackling classification issues. SVC uses the RBF (radial basis function) kernel to determine the optimum line in two dimensions and may also be used to find the line of regression. |
| [65] | XGBoost | XGBoost is used as a fast supervised learning algorithm to achieve accurate and exact cervical cancer categorization citepashraf2022deep. Its regularised learning characteristics make final weight smoothing easier and prevent overfitting. The XGBoost algorithm is based on the minimization of a loss function. |
| [66] | ETC | ETC is a well-known ensemble learning approach for making accurate predictions by combining many decision trees. In contrast to RFt, ETC chooses a subset of the best characteristics at random for each split in the decision tree. This approach produces de-correlated trees, which are less sensitive to individual attributes and more resilient to noise. ETC selects the appropriate feature to partition the data based on the Gini index. It also assesses the significance of traits according to their Gini score. |
| [67] | GNB | GNB is a Bayesian classification method. It presumes that the characteristics are independently dispersed and regularly distributed. In order to determine the probability of each class given the input data, the method first determines the likelihood of a feature instance for each class. |
| [68] | SGD | The LR and SVM underlying ideas are combined in the method known as SGD. SGD is a reliable classifier that utilizes the convex loss function of LR. The OvA (one-versus-all) strategy is used to integrate numerous classifiers, which is particularly helpful for multiclass classification. |

operation is described by algorithm 1, which can be written as follows:

$$\hat{p} = argmax\{\sum_{i}^{n} XGB_i, \sum_{i}^{n} RF_i, \sum_{i}^{n} ETC_i\}. \tag{3}$$

where the following variables offer prediction probabilities for each test sample: $\sum_{i}^{n} XGB_i$, $\sum_{i}^{n} RF_i$, and $\sum_{i}^{n} ETC_i$. After that, as illustrated in Fig 2, the probabilities for each test case employing XGB, RF, and ETC pass through the soft voting.

**Algorithm 1** Ensembling of XGB, RF, and ETC.

```
Input: input data (x, y)_{i=1}^{N}
M_XGB = Trained_XGB
M_RF = Trained_RF
M_ETC = Trained_ETC
1: for i = 1 to M do
2:     if M_XGB ≠ 0 & M_RF ≠ 0 & M_ETC ≠ 0 & training_set ≠ 0 then
```

```
 3:    ProbXGB − Cancer = M_XGB.probability(Cancer − class)
 4:    ProbXGB − Normal = M_XGB.probability(Normal − class)
 5:    ProbRF-Cancer = M_RF.probability(Cancer − class)
 6:    ProbRF-Normal = M_RF.probability(Normal − class)
 7:    ProbETC − Cancer = M_ETC.probability(Cancer − class)
 8:    ProbETC − Normal = M_ETC.probability(Normal − class)
 9:    Decision function = max( 1/N_classifier Σ_classifier
       (Avg_(ProbXGB-Cancer, ProbRF-Cancer, ProbETC-Cancer)
       , (Avg_(ProbXGB-Normal, ProbRF-Normal, ProbETC−Normal)
10: end if
11: Return final label p̂
12: end for
```

The ensemble model combines the predicted probabilities of the two classifiers and selects the final class based on the greatest average probability of a class. The final forecast will be the one with the highest likelihood score, as

$$VC(XGB + RF + ETC) = argmax(g(x)) \tag{4}$$

## Evaluation metrics

The assessment step, which involves evaluating the effectiveness of learning models, is essential for performance analysis. The effectiveness of breast cancer detection models is evaluated using standard assessment metrics including accuracy, precision, recall, and F1 score. Based on the numbers in the confusion matrix, which shows how well the classifier performed on the test data, these parameters are determined. These evaluation parameters are computed using the values of TP (true positive), TN (true negative), FP (false positive), and FN (false negative),

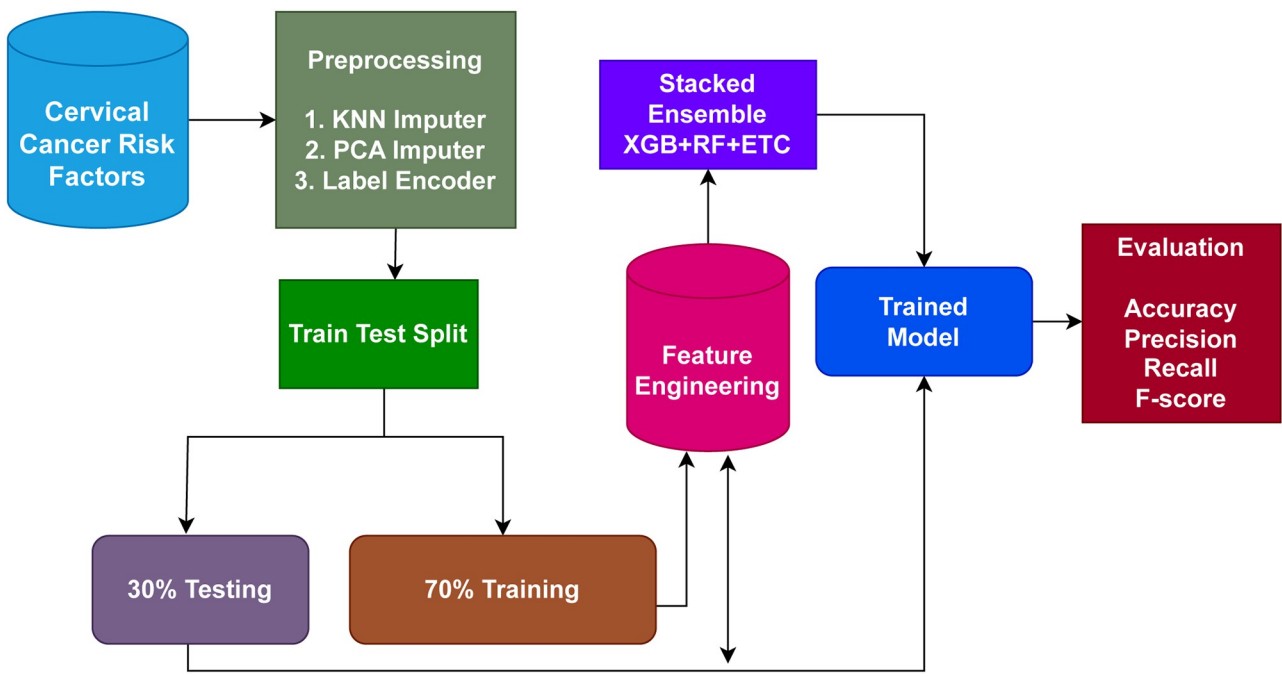

**Fig 1. Workflow diagram of the proposed methodology.**

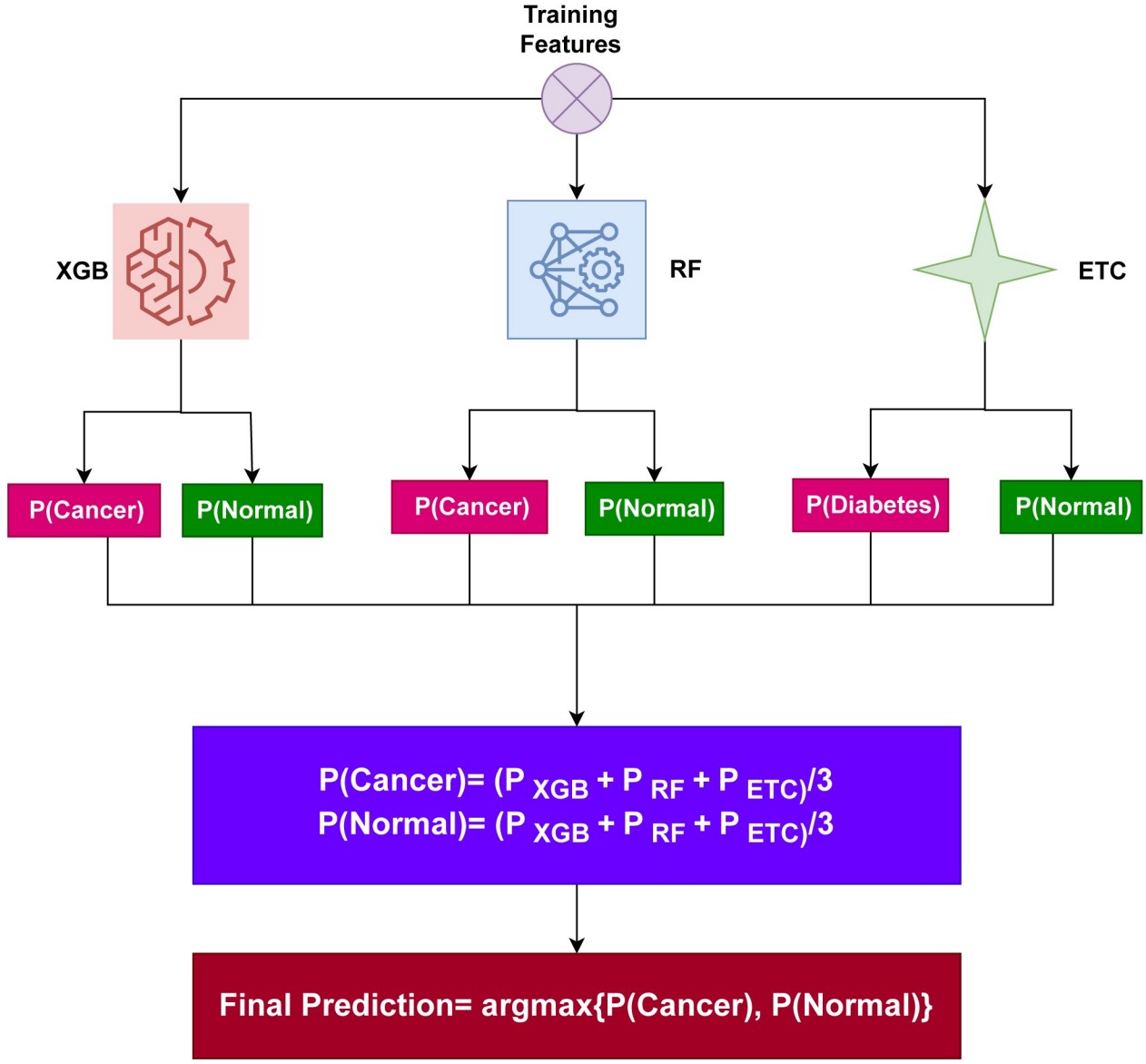

**Fig 2. The architecture of the proposed voting classifier.**

whose values range from 0 (minimum) to 1 (maximum). The accuracy of a classifier's predictions of both positive and negative classes is measured. Mathematically,

$$Accuracy = \frac{TP + TN}{TP + TN + FP + FN} \tag{5}$$

Precision and recall are two regularly used assessment criteria for classifier performance. Precision is defined as the fraction of TP out of all expected positives (TP + FP).

$$Precision = \frac{TP}{TP + FP} \tag{6}$$

Recall, on the other hand, assesses the fraction of TP out of all real positives (TP + FN) and is defined as:

$$Recall = \frac{TP}{TP + FN} \tag{7}$$

The F1 score, which ranges from 0 to 1, is the harmonic mean of accuracy and recall. The F1 score is a well-balanced measure of accuracy and memory. Mathematically, it is as follows:

$$F1 - Score = 2 \times \frac{Precision \times Recall}{Precision + Recall} \tag{8}$$

## Results and discussion

This section presents the results of several classifiers for identifying cervical cancer. The machine learning models are built with Python 3.8 and a Jupyter Notebook, and the tests are run on a system with a 7th-generation Core i7 CPU and Windows 10. The learning models' performance is evaluated using accuracy, precision, recall, and F1 score. Table 3 describes in full the hardware and software specs used in the experiment.

### Results of the ML models with deleted values

The initial stage of the tests involves eliminating missing values from the dataset before applying machine learning models to the updated data. Table 4 summarises the model performance.

According to the results, the RF, ETC, and XGBoost classifiers had the greatest accuracy rates of 71.55%, 72.98%, and 73.41%, respectively. A precision value of 79.25%, a recall value of 80.65%, and an F1 score of 80.11% are likewise displayed by RF, whereas ETC showed a precision value of 80.25%, a recall value of 80.25%, and an F1 score of 80.25%. Similar results are achieved with XGBoost, which received an F1 score of 79.91%, an accuracy value of 79.85%, and a recall value of 79.99%. In contrast, LR performed the least well, with an accuracy of 63.47%, precision of 76.44%, recall of 78.54%, and F1 score of 77.41%.

The proposed VC(XGB+RF+ETC) ensemble model beat all competing learning models with accuracy, precision, recall, and F1 scores of 79.93%, 83.36%, and 84.67%, respectively. Individual machine learning models' overall performance while using the deleted missing value data is poor. The results of the machine learning model with deleted missing value data are represented graphically in Fig 3, which demonstrates that LR has the lowest accuracy scores. The model's performance is impacted by non-constant variation in the data. The performance of other models, aside from RF, ETC, XGBoost, and VC (XGB+RF+ETC), is average, according to the results.

Table 3. Experimental setup for the proposed system.

| Element | Details |
|---|---|
| Language | Python 3.8 |
| OS | 64-bit window 10 |
| RAM | 8GB |
| GPU | Nvidia, 1060, 8GB |
| CPU | Core i7, 7th Gen with 2.8 GHz processor |

**Table 4. Results of the ML models were obtained by deleting missing values from the dataset.**

| Model | Accuracy | Precision | Recall | F1 score |
|---|---|---|---|---|
| LR | 63.47 | 76.44 | 78.54 | 77.41 |
| DT | 67.14 | 77.41 | 79.35 | 78.67 |
| RF | 71.55 | 79.25 | 80.65 | 80.11 |
| SGD | 68.49 | 76.27 | 78.78 | 77.56 |
| ETC | 72.98 | 80.25 | 80.25 | 80.25 |
| XGB | 73.41 | 79.85 | 79.99 | 79.91 |
| SVC | 69.25 | 76.24 | 81.34 | 78.52 |
| GNB | 65.28 | 74.34 | 75.02 | 74.89 |
| Proposed Approach | 79.93 | 83.36 | 85.21 | 84.67 |

## Results of ML models using KNN imputer

The KNN imputer is used for the second set of trials to fill in any missing values in the dataset. It is discovered that certain values are missing after preprocessing the data, which led to the adoption of the KNN imputer to fill in these gaps. The Euclidean distance measure and the mean of the provided values are used to do the imputation. Following that, a number of machine-learning models are trained and tested using the generated dataset. Table 5 shows

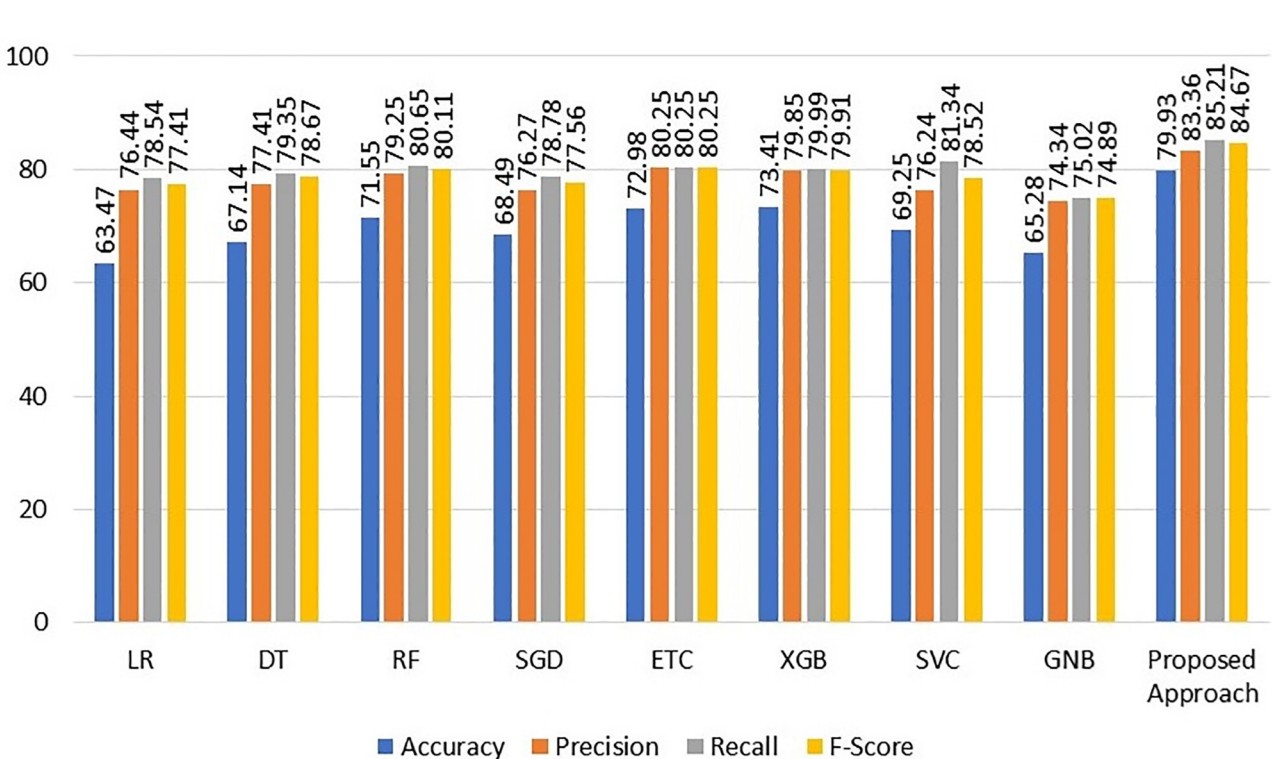

**Fig 3. Results of the ML models after deleting missing values from the dataset.**

**Table 5. Results of the ML models using KNN imputer.**

| Model | Accuracy | Precision | Recall | F1 |
|---|---|---|---|---|
| LR | 73.57 | 86.54 | 88.64 | 87.51 |
| DT | 77.24 | 87.51 | 89.45 | 88.77 |
| RF | 81.65 | 89.35 | 90.88 | 90.31 |
| SGD | 78.69 | 86.41 | 88.83 | 87.86 |
| ETC | 83.10 | 90.33 | 90.33 | 90.33 |
| XGB | 83.52 | 89.74 | 90.25 | 90.01 |
| SVC | 80.54 | 88.42 | 89.43 | 89.25 |
| GNB | 79.82 | 86.43 | 86.20 | 86.98 |
| Proposed Approach | 99.41 | 97.63 | 95.96 | 96.76 |

how different models performed using a dataset enhanced by a KNN imputer. It demonstrates that DT and LR have produced the worst outcomes, with accuracy rates of 73.57% and 77.24% respectively.

The findings show that the accuracy for RF, ETC, and XGBoost is 81.65%, 83.10%, and 83.52%, respectively. The findings are consistent with the prediction that using the KNN imputer to fill in the missing values will enhance the outcomes since machine learning models perform better with the missing values dataset than without. The suggested VC (XGB+RF +ETC) ensemble model has the greatest accuracy rate of any model, at 99.41%. A precision value of 97.63%, a recall value of 95.96%, and an F1 score of 96.76% are also shown for the suggested ensemble model. The accuracy value for the linear model LR, on the other hand, is the lowest at 73.57%. The results of the machine learning models employing the KNN imputer are shown graphically in Fig 4, illustrating how using the KNN imputer enhances the performance of the machine learning models.

### Results of ML models using PCA imputer

The PCA Imputer is used to fill the missing data in the dataset like KNN. It is done using a function called imputePCA. As we can observe from the dataset that there are certain missing values. So, in the third set of experiments, this research work makes use of PCA imputer to fill in the missing values. Following that, a number of machine-learning models are trained and tested using the generated dataset. Table 6 shows how different models performed using PCA imputer. It demonstrates that LR and SGD results are significantly low, with accuracy rates of 69.34% and 74.17% respectively.

The findings show that the accuracy for GNB, ETC, and XGBoost is 83.54%, 81.65%, and 80.19%, respectively. The findings are consistent with the prediction that using the PCA imputer enhances the outcomes since machine learning models perform better with imputing the missing values using the KNN imputer. The suggested VC (XGB+RF+ETC) ensemble model has the greatest accuracy rate of any model, at 94.78%. A precision value of 91.36%, a recall value of 94.62%, and an F1 score of 92.17% are also shown for the suggested ensemble model. The results of the machine learning models employing the PCA imputer is shown graphically in Fig 5, illustrating how using the PCA imputer enhances the performance of the machine learning models.

### Comparison of ML Models by using all techniques

By contrasting the performance of machine learning models with and without the KNN imputer, we are able to determine the KNN imputer's usefulness. The outcomes demonstrated that,

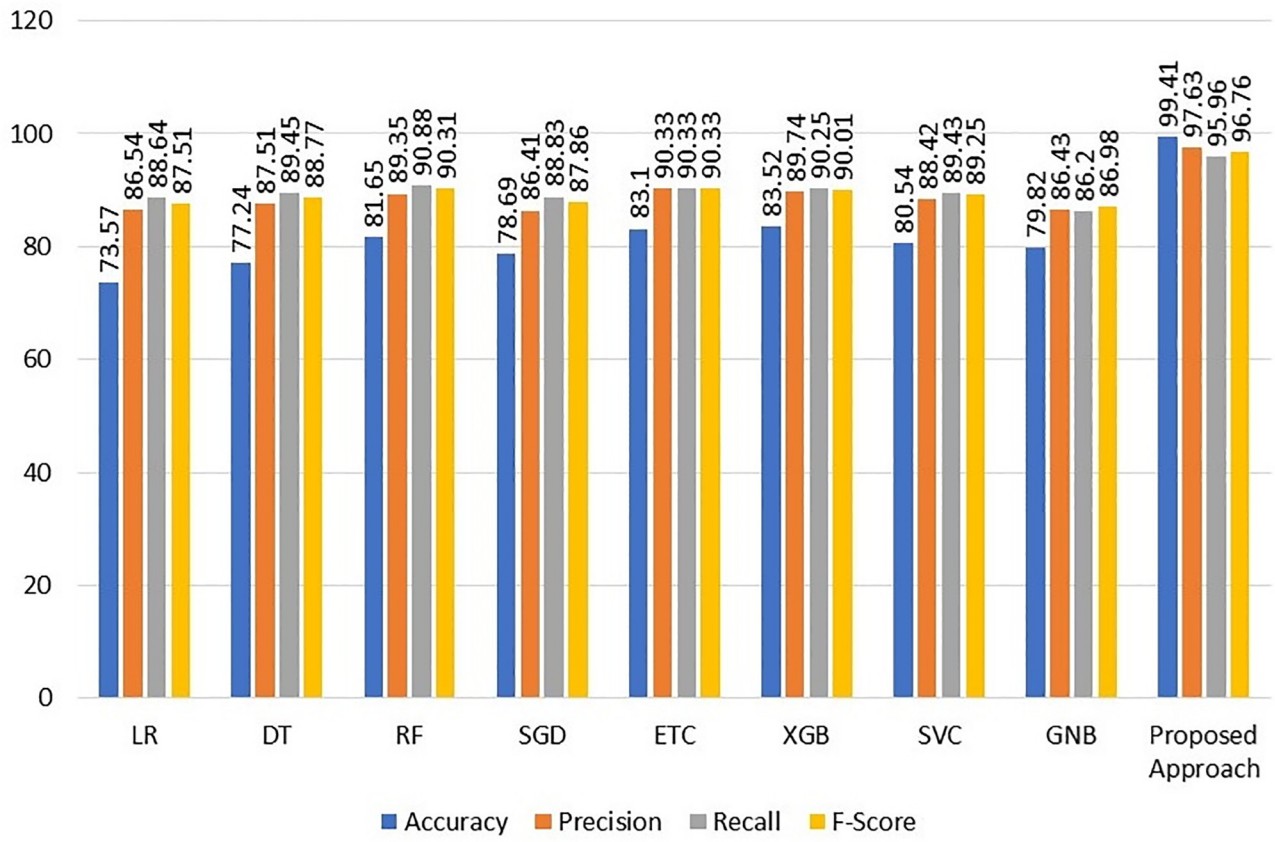

**Fig 4. Results of the learning models using KNN imputer.**

when the KNN imputer is employed in the second experiment instead of utilizing data with removed missing values, the performance of the machine learning models significantly improved. It is simpler to evaluate the performance of the machine learning models by seeing the results in Table 7 for both situations.

Using the removed missing values and the KNN imputed dataset, Fig 6 compares the performance of the machine learning models. The graph demonstrates how using the KNN

**Table 6. Results of the ML models using PCA imputer.**

| Model | Accuracy | Precision | Recall | F1 |
|---|---|---|---|---|
| LR | 69.34 | 82.44 | 86.27 | 84.64 |
| DT | 75.18 | 84.58 | 88.39 | 86.76 |
| RF | 77.15 | 83.27 | 91.23 | 87.37 |
| SGD | 74.17 | 80.28 | 85.15 | 83.27 |
| ETC | 81.65 | 88.47 | 91.29 | 90.27 |
| XGB | 80.19 | 84.85 | 88.07 | 86.67 |
| SVC | 77.67 | 87.14 | 91.72 | 90.27 |
| GNB | 83.54 | 83.49 | 87.19 | 85.76 |
| Proposed Approach | 94.78 | 91.36 | 94.62 | 92.17 |

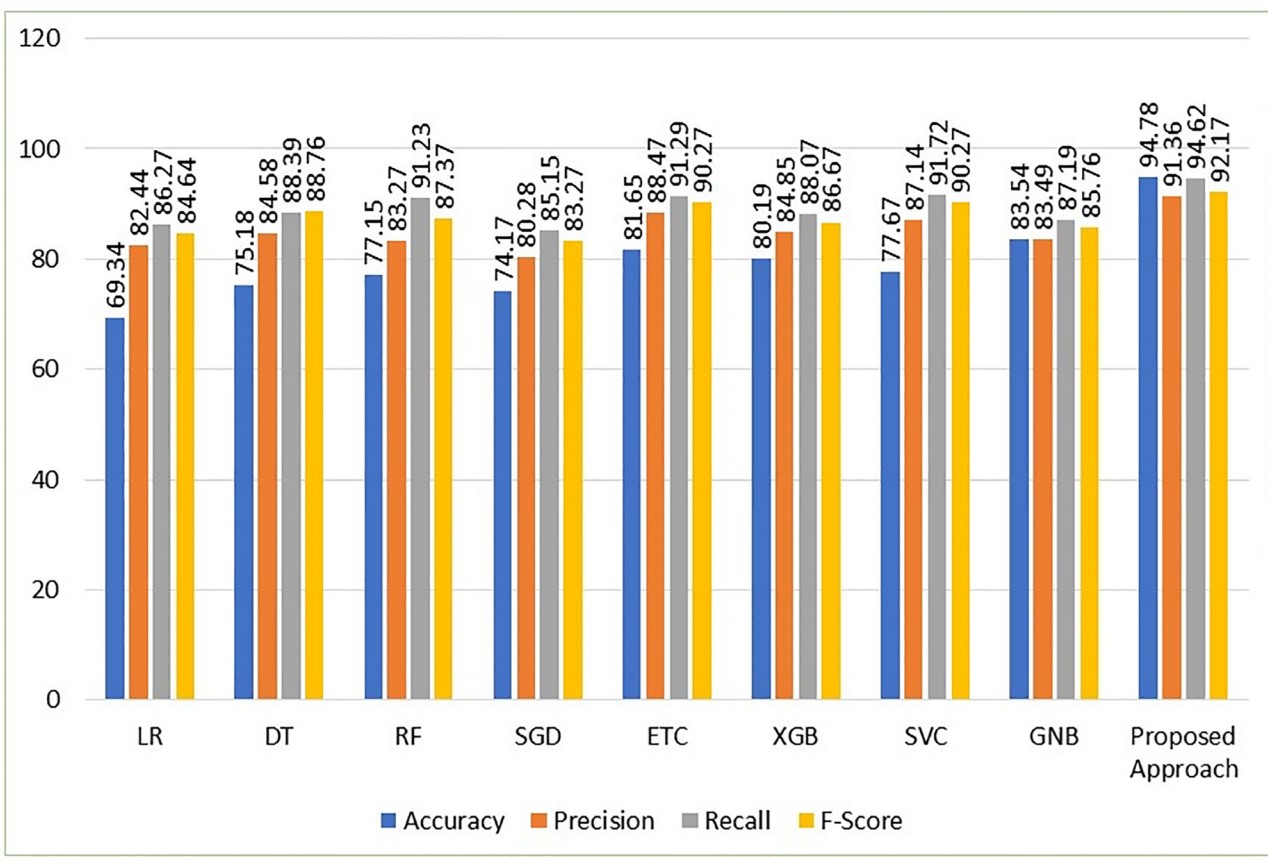

**Fig 5. Results of the learning models using PCA imputer.**

imputer improves the performance of the individual models, leading to greater overall performance from all the machine learning models.

## Limitations of the current work

There are certain dataset restrictions with this study. Only 858 cases with 36 characteristics are included in the current dataset that we used for this study. The constraints of this study include a large number of characteristics and a small number of cases with missing values. This

**Table 7. Accuracy comparison of the ML models.**

| Model | With KNN | Without KNN | With PCA Imputer |
|---|---|---|---|
| LR | 73.57 | 63.47 | 69.34 |
| DT | 77.24 | 67.14 | 75.18 |
| RF | 81.65 | 71.55 | 77.15 |
| SGD | 78.69 | 68.49 | 74.17 |
| ETC | 83.10 | 72.98 | 81.65 |
| XGB | 83.52 | 73.41 | 80.19 |
| SVC | 80.54 | 69.25 | 77.67 |
| GNB | 79.82 | 65.28 | 83.54 |
| Proposed Approach | 99.41 | 79.93 | 94.78 |

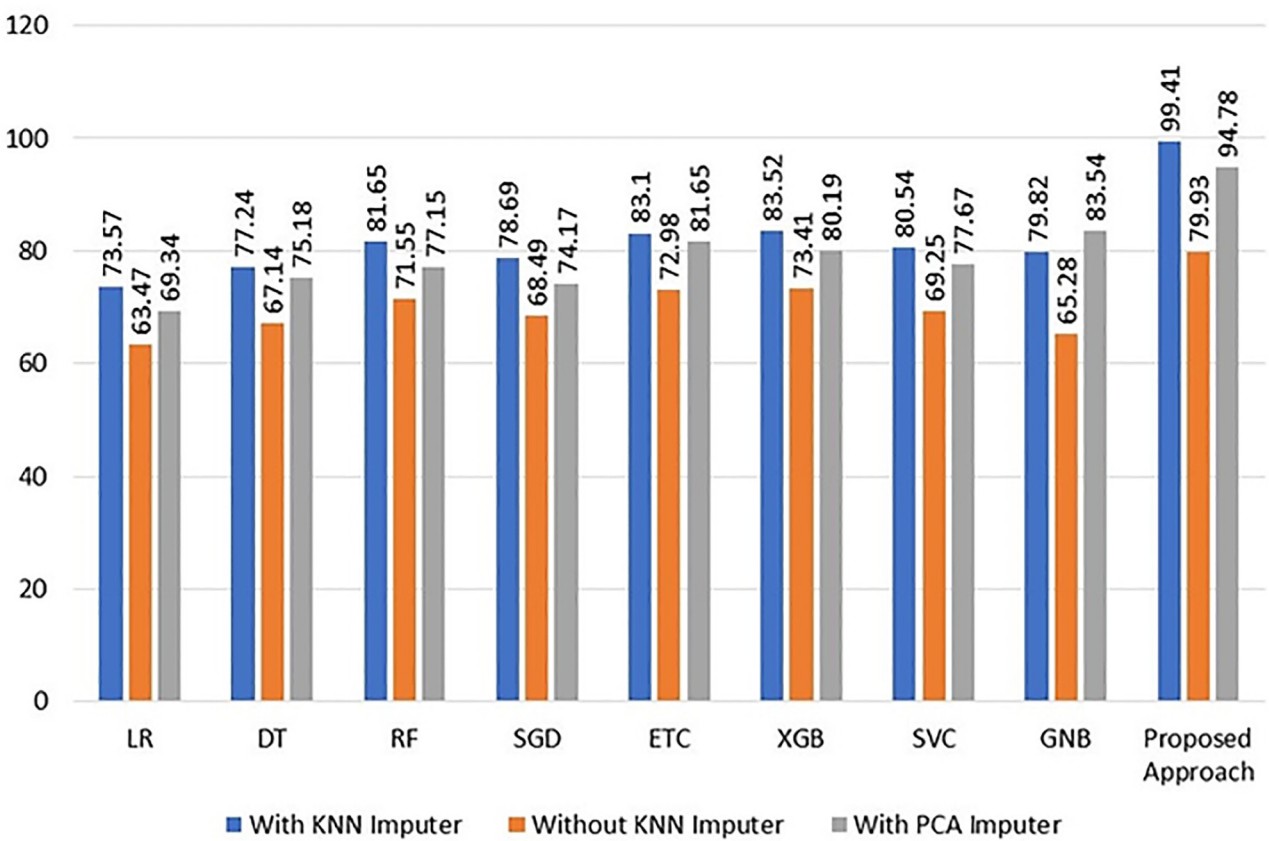

**Fig 6. Performance comparison of the machine learning models regarding the use of KNN imputer.**

research utilizes KNN imputer and PCA imputer to solve the problem. Although the KNN imputer performs admirably in this instance, both still have certain limitations when used with other kinds of datasets. The constraints are:

**KNN imputer.**

- discover the closest neighbours, the KNN imputer takes into account all characteristics. It is possible for some characteristics to add bias to the imputations, though, if they are unimportant or noisy. To solve this problem, careful feature selection or preprocessing may be required.

- Continuous or numerical variables are the main focus of KNN imputer design. When working with categorical variables or mixed-type data, it could not perform effectively. It may be necessary to do further preprocessing processes, such as transforming category data into numerical representations.

- When working with huge datasets or high-dimensional data, KNN imputer might be computationally costly. Finding the closest neighbours and computing imputations might take a lot of time as the number of samples and characteristics grows.

- KNN imputation relies on the closest neighbours to determine local similarity. It could miss more intricate patterns or connections in the data, including non-linear dependencies or global trends.

**PCA imputer.**

- PCA imputer makes the assumption that the data has a multivariate normal distribution, which may not be true for many datasets from the actual world.

- PCA makes the potentially false assumption that the missing data are absent at random. The imputed data may become biased or distorted as a result.

- Due to the fact that PCA employs predicted values rather than actual values and minimizes the number of variables, it may also result in information loss or noise being introduced into the imputed data.

- Due to the way that PCA interprets categorical and ordinal data as continuous variables while ignoring their discrete character, it may not be the best method for these types of variables. If there are too few or too many primary components, this may have an impact on the validity and accuracy of future studies.

## Results of the K-fold cross validation

K-fold cross-validation is used to guarantee the models' dependability. The results of 5-fold cross-validation are presented in Table 8, which shows that the suggested method outperforms previous models in terms of accuracy, precision, recall, and F1 score with a low standard deviation.

## Performance comparison with existing studies

Results are compared with current models to demonstrate how well the suggested model performs in comparison to earlier state-of-the-art models. This study project chooses the three studies that are most closely connected to its goals and uses cutting-edge models to increase accuracy. For instance, [69] earned an accuracy score of 96.06% when it combined SMOTE characteristics with the RF machine learning model to diagnose cervical cancer. The DBSCAN with SMOTETomek and RF machine learning model is utilized in the research [70] to reach the maximum accuracy score of 97.72%. Similar to this, the [71] uses RFE (recursive feature elimination) with SMOTETomek and reports an accuracy score of 98.81%. The performance comparison between the suggested and existing research is shown in Table 9. Results show improved functionality of the proposed model.

## Limitations of proposed model with KNN imputation techniques

K-Nearest Neighbors (KNN) imputation is a popular method for filling in missing values in datasets, but it has several limitations:

**Table 8. 5-fold- cross-validation results for the proposed approach.**

| Model | Accuracy | Precision | Sensitivity | F1 score |
|---|---|---|---|---|
| 1st fold | 98.52 | 95.13 | 94.61 | 95.12 |
| 2nd fold | 98.25 | 96.34 | 97.74 | 97.23 |
| 3rd fold | 99.64 | 95.67 | 95.98 | 95.81 |
| 4th fold | 99.08 | 94.78 | 94.99 | 94.85 |
| 5th fold | 99.98 | 97.15 | 95.86 | 96.33 |
| **Average** | **99.09** | **95.82** | **95.84** | **95.90** |
| **Std. Deviation** | **± 0.70** | **± 0.85** | **± 1.08** | **± 0.85** |

**Table 9. Performance comparison with state-of-the-art studies.**

| Ref. | Technique | Features | Accuracy |
|---|---|---|---|
| [69] | SMOTE with RF | 30 | 96.06% |
| [69] | SMOTE with RF and RFE | 18 | 95.87% |
| [69] | SMOTE with RF and PCA | 11 | 95.74% |
| [70] | DBSCAN with SMOTETomek and RF | 10 | 97.72% |
| [70] | DBSCAN with SMOTETomek and RF | 10 | 97.22% |
| [70] | iForest with SMOTETomek and RF | 10 | 97.50% |
| [70] | iForest with SMOTE and RF | 10 | 97.58% |
| [71] | RFE with DT | 20 | 97.65% |
| [71] | Lasso with DT | 10 | 96.47% |
| [71] | RFE with SMOTETomek and DT | 20 | 98.82% |
| [71] | Lasso with SMOTETomek and DT | 10 | 92.92% |
| **Proposed Approach** | **Stacked Ensemble VC (XGB+RF+ETC) with KNN Imputer** | **30** | **99.41%** |

- Sensitivity to the Value of K: The performance of KNN imputation is sensitive to the choice of the number of nearest neighbors (K). A small K may result in noise being introduced into the imputed values, while a large K may lead to over smoothing and loss of important patterns in the data.

- Curse of Dimensionality: In high-dimensional spaces, the notion of "nearest neighbors" becomes less meaningful because all data points tend to be relatively far apart. This makes KNN less effective in such settings.

- Handling Categorical Data: KNN is primarily designed for numerical data and may not work well with categorical features. Special techniques, like Hamming distance or Gower distance, need to be used for mixed data types.

- Interpretability: KNN imputed values may not be as interpretable as those obtained through other imputation methods. The imputed values are based on the values of neighboring data points rather than a model-based approach.

- Noisy Data: KNN is sensitive to noisy data because it relies on distance calculations. Outliers or noisy data points can have a disproportionate impact on the imputed values.

Impact of Outliers: Outliers in the dataset can significantly influence KNN imputation. If there are outliers, the nearest neighbors chosen for imputation may not be representative of the local data distribution, leading to inaccurate imputed values.

Equal Weighting of Neighbors: KNN imputation assumes that all K nearest neighbors contribute equally to imputing a missing value. In reality, some neighbors may be more similar to the missing instance than others, but KNN treats them all equally.

Missing Data Patterns: KNN imputation assumes that missing values are Missing Completely at Random (MCAR) or Missing at Random (MAR). If data is Missing Not at Random (MNAR), KNN imputation can introduce bias.

Scalability: For large datasets, the memory and computational requirements of KNN can become a significant limitation.

## Conclusions

Cervical cancer is still a major worldwide health issue, especially in developing nations with poor access to healthcare and preventative measures. But improvements in screening methods

have helped with early detection and raised survival rates. However, the risk of mortality associated with cervical cancer can be substantially mitigated through early detection and timely treatment. Machine learning techniques have been demonstrated to provide enhanced accuracy in the detection of cervical cancer in this domain.

This research study introduces a two-part framework aimed at effectively identifying cervical cancer in patients. In the initial phase, the dataset undergoes normalization using the KNN imputer approach. Subsequently, in the second phase, the stacked ensemble voting classifier (XGB+RF+ETC) model is utilized. The application of ensemble models proves to be a robust choice for the early detection of cervical cancer, as evidenced by the highly accurate results of 99.41%. Additionally, this study explores the implementation of the PCA imputer approach. The superiority of the proposed model is further evidenced through a comparison with other state-of-the-art models. As part of future work, this research aims to develop a layered combination of deep learning and machine learning models. The objective is to enhance the model's performance on datasets with higher dimensions and achieve robust and generalized results to save women's health.

## Author Contributions

**Conceptualization:** Turki Aljrees.

**Data curation:** Turki Aljrees.

**Formal analysis:** Turki Aljrees.

**Funding acquisition:** Turki Aljrees.

**Investigation:** Turki Aljrees.

**Methodology:** Turki Aljrees.

**Project administration:** Turki Aljrees.

**Resources:** Turki Aljrees.

**Software:** Turki Aljrees.

**Supervision:** Turki Aljrees.

**Validation:** Turki Aljrees.

**Visualization:** Turki Aljrees.

**Writing – original draft:** Turki Aljrees.

**Writing – review & editing:** Turki Aljrees.

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
