## [Decision Letter · Decision Letter 0]

29 Aug 2023

PONE-D-23-24373Improving Prediction of Cervical Cancer Using KNN Imputer and Multi-Model Ensemble LearningPLOS ONE

Dear Dr. Aljrees,

Thank you for submitting your manuscript to PLOS ONE. After careful consideration, we feel that it has merit but does not fully meet PLOS ONE’s publication criteria as it currently stands. Therefore, we invite you to submit a revised version of the manuscript that addresses the points raised during the review process.

We look forward to receiving your revised manuscript.

Kind regards,

Saddam Hussain Khan

Academic Editor

PLOS ONE

Journal Requirements:

   "The author received no specific funding for this work. He will manage the APC himself."

Reviewers' comments:

Reviewer's Responses to Questions

**Comments to the Author**

1. Is the manuscript technically sound, and do the data support the conclusions?

Reviewer #1: Yes

Reviewer #2: Yes

2. Has the statistical analysis been performed appropriately and rigorously? 

Reviewer #1: No

Reviewer #2: No

3. Have the authors made all data underlying the findings in their manuscript fully available?

Reviewer #1: Yes

Reviewer #2: Yes

4. Is the manuscript presented in an intelligible fashion and written in standard English?

Reviewer #1: Yes

Reviewer #2: No

5. Review Comments to the Author

Reviewer #1: The author presents an interesting idea of utilizing ML techniques for the cervical cancer survival prediction. In their work, they utilized KNN and PCA imputer to deal with the missing values. However, there are some concerns which need to be addressed before publication.

1- Introduction section need revision, it is recommended to present one idea in one paragraph. As in paragraph one, the author introduces cervical cancer and its statistics, which are again repeated in paragraph 3. In addition, provide some information about some common methods used to detect cervical cancer survival rate.

2- Reconsider the Related work section to incorporate recently published works for cervical cancer survival prediction. 3- 3- 3- Clearly mention the research gap.

4- Highlight the motivation and rationale of the study.

5-Improve images to make them more clear.

6- Empirical analysis of the proposed methodology is missing.

7- Provide ablation study to show the significance of each part of the proposed methodology.

8- Why PCA was used and how it only selected the important features.

9- Overall formatting needs reconsideration, at places text seems be unaligned, and indentation is not consistent.

10- Number of grammatical and typo mistakes are present. Thoroughly revise the paper

Reviewer #2: The author introduces an intriguing concept of employing machine learning techniques for cervical cancer survival prediction. In their study, KNN and PCA imputer were employed to address missing values. However, several concerns must be addressed before publication.

1. The abstract missed the novelty, Challenges, validation techniques, and some suggestion. Impact on social and Clinical.

2. The introduction section requires revision, with a suggestion to present one idea per paragraph. For instance, the author discusses cervical cancer and its statistics in both the first and third paragraphs. Furthermore, it would be beneficial to include details about common methods used for detecting cervical cancer survival rates.

3. Revisiting the Related Work section would be advantageous to incorporate recently published research on cervical cancer survival prediction.

4. Clearly outlining the research gap would enhance the paper's clarity.

5. The proposed work is based on KNN/PCA which is state of the art algorithm, however, a lot of work exists on the proposed techniques. Please, mention your novelty in detail, rational, and impact of your techniques.

6. Please, provide the detail ablation study in terms of results, and computational complexity.

7. PR and ROC curve may be employed. Moreover, feature space visualization may require.

8. Please, give limitation of previous work and critical research gap.

9. Please, provide the detail ablation study in terms of results, and computational complexity.

10. Emphasizing the motivation and rationale behind the study is important.

11. Images could be improved to enhance their clarity and comprehension.

12. The paper lacks empirical analysis of the proposed methodology.

13. The literature review can be strengthened by refereeing the articles

• Zafar, Muhammad Mohsin, Zunaira Rauf, Anabia Sohail, Abdul Rehman Khan, Muhammad Obaidullah, Saddam Hussain Khan, Yeon Soo Lee, and Asifullah Khan. "Detection of tumour infiltrating lymphocytes in CD3 and CD8 stained histopathological images using a two-phase deep CNN." Photodiagnosis and Photodynamic Therapy 37 (2022): 102676.

• Zahoor, M. M., & Khan, S. H. (2022). Brain tumor MRI Classification using a Novel Deep Residual and Regional CNN. arXiv preprint arXiv:2211.16571.

• Khan, S. H., Sohail, A., Zafar, M. M., & Khan, A. (2021). Coronavirus disease analysis using chest X-ray images and a novel deep convolutional neural network. Photodiagnosis and Photodynamic Therapy, 35, 102473.

• Zahoor, M.M.; Qureshi, S.A.; Bibi, S.; Khan, S.H.; Khan, A.; Ghafoor, U.; Bhutta, M.R. A New Deep Hybrid Boosted and Ensemble Learning-Based Brain Tumor Analysis Using MRI. Sensors 2022, 22, 2726. https://doi.org/10.3390/s22072726

• Rauf, Zunaira, Anabia Sohail, Saddam Hussain Khan, Asifullah Khan, Jeonghwan Gwak, and Muhammad Maqbool. "Attention-guided multi-scale deep object detection framework for lymphocyte analysis in IHC histological images." Microscopy 72, no. 1 (2023): 27-42.

• Khan, Saddam Hussain, Najmus Saher Shah, Rabia Nuzhat, Abdul Majid, Hani Alquhayz, and Asifullah Khan. "Malaria parasite classification framework using a novel channel squeezed and boosted CNN." Microscopy 71, no. 5 (2022): 271-282.

6. PLOS authors have the option to publish the peer review history of their article (what does this mean?). If published, this will include your full peer review and any attached files.

Reviewer #1: No

Reviewer #2: No

---

## [Author Response · Author response to Decision Letter 0]

22 Oct 2023

The response is attached as a separate file named as response to reviewers. (PDF)

---

## [Decision Letter · Decision Letter 1]

21 Nov 2023

PONE-D-23-24373R1Improving Prediction of Cervical Cancer Using KNN Imputer and Multi-Model Ensemble LearningPLOS ONE

Dear Dr. Aljrees,

Thank you for submitting your manuscript to PLOS ONE. After careful consideration, we feel that it has merit but does not fully meet PLOS ONE’s publication criteria as it currently stands. Therefore, we invite you to submit a revised version of the manuscript that addresses the points raised during the review process.

We look forward to receiving your revised manuscript.

Kind regards,

Saddam Hussain Khan

Academic Editor

PLOS ONE

Journal Requirements:

Additional Editor Comments:

Author has fulfilled the concerns, but I suggest to include more recent papers like

"10.1007/s10462-023-10595-0" , "10.1016/j.imed.2022.07.002", "10.1038/s41598-023-40581-z".

Secondly, reconsider the placement of Figure 1 in the paper.

Also improve the figure captions to be more descriptive.

Please, incudes articles for critical research gap and future research

https://arxiv.org/abs/2212.02477

https://link.springer.com/article/10.1007/s11042-022-14061-x

https://doi.org/10.1016/j.eswa.2023.120477

Reviewers' comments:

Reviewer's Responses to Questions

**Comments to the Author**

1. If the authors have adequately addressed your comments raised in a previous round of review and you feel that this manuscript is now acceptable for publication, you may indicate that here to bypass the “Comments to the Author” section, enter your conflict of interest statement in the “Confidential to Editor” section, and submit your "Accept" recommendation.

Reviewer #1: All comments have been addressed

Reviewer #2: All comments have been addressed

2. Is the manuscript technically sound, and do the data support the conclusions?

Reviewer #1: Yes

Reviewer #2: Partly

3. Has the statistical analysis been performed appropriately and rigorously? 

Reviewer #1: N/A

Reviewer #2: I Don't Know

4. Have the authors made all data underlying the findings in their manuscript fully available?

Reviewer #1: Yes

Reviewer #2: Yes

5. Is the manuscript presented in an intelligible fashion and written in standard English?

Reviewer #1: Yes

Reviewer #2: Yes

6. Review Comments to the Author

Reviewer #1: Author has fulfilled the concerns, but I suggest to include more recent papers like

"10.1007/s10462-023-10595-0" , "10.1016/j.imed.2022.07.002", "10.1038/s41598-023-40581-z".

Secondly, reconsider the placement of Figure 1 in the paper.

Also improve the figure captions to be more descriptive.

Reviewer #2: (No Response)

7. PLOS authors have the option to publish the peer review history of their article (what does this mean?). If published, this will include your full peer review and any attached files.

Reviewer #1: No

Reviewer #2: No

---

## [Author Response · Author response to Decision Letter 1]

23 Nov 2023

We have added a separate response to reviewer PDF file.

---

## [Editor Report · Decision Letter 2]

24 Nov 2023

Improving Prediction of Cervical Cancer Using KNN Imputer and Multi-Model Ensemble Learning

PONE-D-23-24373R2

Dear Dr. Aljrees,

We’re pleased to inform you that your manuscript has been judged scientifically suitable for publication and will be formally accepted for publication once it meets all outstanding technical requirements.

Kind regards,

Saddam Hussain Khan

Academic Editor

PLOS ONE
---

## [Editor Report · Acceptance letter]

30 Nov 2023

PONE-D-23-24373R2 

Improving Prediction of Cervical Cancer Using KNN Imputer and Multi-Model Ensemble Learning 

Dear Dr. Aljrees:

I'm pleased to inform you that your manuscript has been deemed suitable for publication in PLOS ONE. Congratulations! Your manuscript is now with our production department. 

Kind regards, 

on behalf of

Dr. Saddam Hussain Khan 

Academic Editor

PLOS ONE